# Stress and Strain among Seafarers Related to the Occupational Groups

**DOI:** 10.3390/ijerph16071153

**Published:** 2019-03-30

**Authors:** Marcus Oldenburg, Hans-Joachim Jensen

**Affiliations:** 1Institute for Occupational and Maritime Medicine Hamburg (ZfAM), University Medical Center Hamburg-Eppendorf (UKE), 20459 Hamburg, Germany; 2Institute for Occupational and Maritime Medicine Hamburg (ZfAM), Department of Maritime Medicine, 20459 Hamburg, Germany; hans_joachim-jensen@t-online.de

**Keywords:** maritime, occupational groups, stress, strain, seafarer

## Abstract

The present study analyses whether the stress and strain experienced by seafarers differ between the various occupational groups on board container ships. In a maritime field study, 323 sailors on 22 container ships were asked to complete a questionnaire and were biometrically surveyed. In addition, a survey of energy expenditure and heart rate (variability) was carried out with the SenseWear^®^ armband monitor and the Polar RS800 watch, respectively. The activity data objectively collected by the armband monitor showed an average sleep duration of 5.0 h per day, with particularly short sleep episodes amongst nautical officers. This occupational group also significantly more frequently reported sleep deficits (67%). The highest work-related energy expenditure per day was among the deck ratings (801 kcal), followed by the engine room personnel (777 kcal), and finally the nautical officers (568 kcal). The last-mentioned group, who were also the most likely to experience mental stress in the workplace, had the lowest heart rate variability compared to the other occupational groups. The average working time was the only stress parameter that correlated significantly negatively with the heart rate variability (*r* = −0.387; *p* = 0.002). Overall, job-related stressors of seafarers on board should be objectified in further studies and occupational group-specific health promotion programmes should be developed.

## 1. Introduction

The largest occupational group on board a ship is the deck crew, which is made up of nautical officers and deck ratings. The nautical officers, including the captain, are responsible for, among other things, navigation, planning and organization of the loading and unloading operations and for the ship’s command [1]. Thus, they have a high level of responsibility for personnel and material. Deck ratings acting as subordinate workers on board carry out tasks outside the engine room, such as mooring and unmooring the ship, loading and unloading cargo, bridge or gangway monitoring, serving as lookouts on the bridge, or undertaking repair and maintenance work in the deck area [1].

The employees in the engine room must be distinguished from the deck personnel. They are exposed to noise, vibration and heat or pollutants during their working hours [2]. The exposure of the technical officers and the ratings working in the engine room are very similar, so that these can be summarized as engine room personnel. However, it should be noted that technical officers are stressed by their higher level of responsibility, especially when the engine alarm is triggered. Another specificity of this group is that the engine room personnel are normally employed in a day shift system and, in contrast to deck crews, are generally not subjected to a watch system [3]. Maintenance and repairs in the engine room are often carried out jointly by the technical officers and the engine room ratings.

In view of the different fields of activity and the associated responsibilities, it can be assumed that the stress and strain of the various occupational groups on board differ considerably [4]. Thus, the required presence of the captain and partly also of other nautical officers on the bridge during a river passage with crowded ship lanes, but also during the ship’s port stay and the subsequent contact with the port authorities, for example, mean a distinct task accumulation [5]. Due to considerable personnel costs involved in a maritime field survey, there are hardly any comprehensive studies about job-related burden and psychophysical strain on board ships [6,7,8]. Therefore, this study aims to assess the stress and psychophysical strain of seafarers in a comprehensive survey using objective methods. In this on-board investigation, a distinction is made between the three occupational groups of nautical officers, deck ratings and engine room personnel. They not only have different tasks, but differences in their average working hours are also assumed.

## 2. Materials and Methods 

Two to four scientific researchers accompanied 22 sea voyages on container ships under German management in the North/Baltic Sea area (including the English Channel) or in a comparable coastal operation with a high port frequency. Over a period of three consecutive days, the study participants recorded their working time, leisure time, lying/sleeping time and times spent on sport as close to the minute and as continuously as possible. At the same time, a continuous objective measurement of strain was carried out on board using the SenseWear^®^ armband monitor (SMT medical GmbH & Co, Wuerzburg, Germany) and the Polar RS800 watch (Polar Electro Oy, Kempele, Finland), throughout the survey period. The SenseWear^®^ armband monitors objectively assess the physical activity (lying or sleeping time as well as the number of steps of the wearers) and calorie expenditure of the sailors. The armband monitor has already been tested and successfully used as an activity measuring system in numerous studies [9,10,11]; it has proved to be superior in comparison to other activity monitors [12].

The Polar RS800 watch provides data on heart rate and heart rate variability (standard deviation of normal to normal R-R intervals, where R is the peak of a QRS complex (SDNN)). The Polar watch has also already been applied in various studies as a suitable measure to assess the mentioned cardiac parameters [13,14]. Both the average heart rate over a period of 24 h and (adapted to the respective four-hour work shift of the nautical officers) the portion of the heart rate above the cardiac continuous performance limit during a four-hour working period were determined. Based on the recommendations of the Employer’s Liability Insurance Association of the Construction Industry (“Berufsgenossenschaft Bau”), this limit was defined as a heart rate of 110 beats per minute (min^−1^) for pragmatic reasons. In the present study, both heart rate and energy expenditure were measured continuously from 236 seafarers over an average study period of 2.8 days (between 2.5–6 days).

Furthermore, the sailors were asked about their subjectively experienced stress due to job-related physical or mental impacts, their maximum working hours, their sleep deficit and the average frequency of sleep interruptions. In accordance with ISO 10075-1:2017 [15], this study defined stress as the entirety of measurable external influences. All questions used were derived from a previously published study on seafaring stressors [2]. They were developed for a multicultural crew with different English-speaking and educational backgrounds. The answer options about physical and mental stress were structured as a five-item Likert scale. 

### 2.1. Study Population

Of the total 365 seafarers on the 22 ships, 323 exclusively male sailors took part in this survey (a participation rate of 88.5%). The study sample was composed of 155 Europeans (48.0%) and 168 Southeast Asians (52.0%; mostly Filipinos). Stratification by occupational groups included 122 officers (37.8%, 67 nautical and 55 technical officers) and 201 ratings (62.2%), representing the usual ratio of number of officers to number of ratings on board container ships. Seafarers were further classified into the three occupational groups: “Nautical officers”, “Deck ratings” and “Engine room personnel” (67, 158 and 98 subjects), as described above. The average age was 38.3 (SD 11.8) years, whereas the engine room personnel were somewhat older (Table 1).

While European seafarers in this study—according to the usual international standard in seafaring—predominantly hold officer’s ranks (nautical and technical officers), Southeast Asians, especially in the deck department, comprised a significantly higher number of ratings.

Participation in this study was completely voluntary and the collected data was pseudonymized. All participants gave their written informed consent for inclusion before taking part in this study. The study was conducted in accordance with the Declaration of Helsinki, and the protocol was approved by the Ethics Committee of the Hamburg Medical Association (No. PV4395).

### 2.2. Statistics

Statistical analysis was performed with SPSS (version 25, IBM Corporation, New York, USA). Continuous, symmetrically distributed variables were expressed as mean (±standard deviation (SD) or ±standard error (SE)). The Pearson chi-square test (Chi^2^ test) was used to compare frequencies between groups. Continuous variables were compared with the Kruskal–Wallis-test. Furthermore, the mixed model with different variances proved most suitable. This statistical model contains both fixed effects and random effects and is used for repeated measurements on the same statistical unit. In the mixed models, the sailors were modelled as a random effect. The differences between certain means were calculated and the estimated marginal means were given as estimates for the predicted means of the cells in the model. 

In a following step, the age adjustment variable was added. Moreover, there was an additional adjustment for the parameters of length of stay on board at the time of the examination, average sleeping and working time, number of terminals called at, subjective stress on board due to physical or mental requirements as well as the objective recording of the sea conditions from the ship’s journal. Additionally, the Spearman’s correlation was conducted to assess the dependence between the rankings of two variables. All reported *p*-values were two-tailed and a *p*-value < 0.05 was considered statistically significant.

## 3. Results

### 3.1. Questionnaire Survey

Physical or mental stress on board the current ship was reported by around 65% of the respondents in the total study sample. There were significant differences between the rankings: 74% of the ratings more often felt physically stressed and more than 86% of the nautical officers were more often mentally stressed. According to the anamnestic data, the average working time on board was 9.5 h, with the watchkeeping nautical officers recording significantly longer working hours than the other occupational groups (Table 2). The maximum working hours among the officers were particularly high.

Furthermore, over 67% of the nautical officers and 56% of the deck ratings reported having a sleep deficit. Accordingly, nautical watch officers experienced the most frequent (occupational or non-occupational) sleep interruptions of a planned bedtime, followed by the deck ratings (Table 2).

### 3.2. Daily Log/Armband Monitor

With the daily log, seafarers continuously documented their current level of activity (work, leisure, sleep and sports) as accurately as possible over the course of several days. Since the activity levels were collected in parallel by all participating seafarers, it was possible to record over 1390 man-days in this way. Based on the total sample, it was established that the daily routine on board consisted of 39.1% working time, 28.3% leisure time, 32.3% sleeping time and 0.4% leisure time for sporting activities (Table 3).

The stratification by occupational groups showed that, compared to the other occupational groups, the nautical officers had a significantly longer working time on average and a much shorter lying and sleeping time. The technical officers had the longest leisure time and this group also documented a higher level of sporting activity (Table 3).

The activity data collected objectively with the armband showed an average lying time of 6.1 h with a sleep duration of 5.0 h per day and a wearing duration of approximately 92.0% within the recorded period of observation. Notably, the nautical officers had a shorter lying and sleeping duration according to the measurements with the armband monitor. Over 90% of the active time declared in the daily log (work, leisure, sport) was recognized by the monitor as non-lying or sleeping time.

The average number of steps differed markedly among the occupational groups, with the crew ratings (especially on deck) recording the highest and the (nautical) officers recording the lowest number of steps (Table 3).

The total energy conversion was 3394 kcal. Work-related energy expenditure and METs (metabolic equivalent of tasks) per day were significantly different between crew groups (*p* < 0.001), with the highest values for the deck ratings, followed by the engine room crew, and finally the nautical officers (Table 4). 

Altogether, there were no differences in the average heart rate between the occupational groups. Higher loads above the endurance limit (>110 beats per minute in relation to a 4-h work shift) occurred less frequently among nautical officers. With regard to heart rate variability (SDNN as a parameter of long-term psychophysical stress), significantly lower values were found among nautical officers (Table 4).

The average heart rate was 82 beats min^−1^. After adjustment for the length of stay on board at the time of the examination and the average sleeping time, it was found that these parameters had no appreciable influence on the individual strain. The only exception was the average working time, which was negatively associated with heart rate variability (*r* = −0.387; *p* = 0.002). The total energy expenditure was positively associated with the frequency of port calls (*r* = 0.293; *p* = 0.036).

Additionally, the sea conditions and the number of terminals called at did not significantly correlate with the heart rate or the overall variability. However, in an isolated analysis of working hours, reduced heart rate variability was found with increasing numbers of terminals (*r* = 0.327; *p* = 0.032). A differentiation by activity levels showed that both heart rate and variability only differed significantly between occupational groups during working hours, with the highest heart rates in the deck ratings followed by the engine room personnel, and the lowest degree among nautical officers.

## 4. Discussion

The anamnestic data of the seafarers on their average working time reflect the documented working time lengths in the current daily log well—as an expression of a high representativeness of the data currently collected on board. Assuming an even distribution of work during the voyage, a daily workload of 39.1% corresponds to an average workday of 9.3 h. According to the daily log, the average time spent lying down/sleeping was 7.8 h per day.

An average working time of 9.3 h on board equates to a working week of just under 66 h per week, as a ship is in continuous operation, especially if it operates in a coastal area. This means that even at weekends and on public holidays work must be done—especially if the ship is in port and the mooring fees for the shipping company are very high [16]. The average working time correlated negatively with the SDNN in the total sample (i.e., longer working time resulted in an increasing restriction of the heart rate variability). This association highlights that the onboard psychophysical stress is essentially determined by the duration of the daily working hours.

This study also demonstrated reduced heart rate variability during working hours in addition to an expectedly significant increase in total energy expenditure as the number of terminal calls increased. This suggests that the psychophysical stress is also significantly influenced by the frequency of port turnover [17]. By contrast, the sea conditions and thus the ship’s movements in this study had no measurably significant effect on the experienced subjective or objective strain.

Furthermore, it became apparent that both the heart rate and its variability only differed significantly between the crew groups during working hours. Since the strain parameters were similar between the occupational groups during leisure and sleep time, a similar compensation ability of the crew groups outside of their working hours is to be assumed. After adjustment for the length of stay on board and the average sleep time, it was found that none of these parameters exerted a significant influence on the individual strain parameters. Thus, it can be assumed that the strain measured in this study was primarily related to the acute situation on board and that it was essentially independent of long-term effects.

Considering the 12-month operational time of the crews on board, such high workloads can lead to chronic exhaustion [18,19]. The manifestation of chronic diseases among seafarers is also abundantly described [20,21,22]. Therefore, from the perspective of preventive medicine, a limitation of at least the length of assignment on board seems to be required [23,24].

It is well known that one of the most urgent health problems among seafarers is chronic fatigue [6,25,26,27]. More than 80% of shipwrecks are due to human error, with fatigue often being considered causal [28,29]. In the present study, an average time of only 6.1 h per day spent lying down was determined using the armband monitor, with an effective sleep duration of 5.0 h. In contrast to official working time records, which are difficult to verify [30], this study shows for the first time a significant lack of sleep among seafarers on the basis of objectively collected onboard working time documentation validated by onboard examiners. The intervals recorded in the daily log as lying/sleeping time were detected with the armband monitor at 70.5% as lying/sleeping time (sensitivity). By contrast, the armband monitor interpreted 93.3% of the active time (work, leisure and sport) declared according to the daily log as non-lying or sleeping time (specificity). In particular, this high specificity suggests that the armband monitor is a suitable measuring instrument for the recording of sleeping and lying times, as has already been shown in other studies [31,32].

This lack of sleep was also subjectively confirmed by nearly 55% of the sailors surveyed, especially nautical officers. Correspondingly, the latter also reported significantly more frequently that they were repeatedly disturbed in their daily sleeping patterns. It should be noted that nautical officers are usually deployed in a 4/8 watch system, which means that a 4-h working shift is followed by 8 h off. Experience has shown that sleep periods are often distributed in the two 8-h recreational periods per day, such that the 5 h of sleep per day are sometimes not taken in one stretch [33].

In addition to the nautical officers, the deck ratings are often engaged in a watch system with often chronobiologically unfavourable and irregular operations (gangway watch or look-out). This also explains the shorter sleep duration and the more frequent sleep interruptions of this occupational group. In contrast, sleep disruptions occur less frequently among the engine room personnel, who are often active in a regular diurnal rhythm.

According to the present questionnaire, about 65% of the seafarers on board were subjectively affected by psychophysical stress, with the nautical officers more frequently feeling mentally stressed. The high mental stress in the nautical officers was possibly also expressed in a significantly lower heart rate variability (SDNN) compared to the other occupational groups. This high stress level results, on the one hand, from the large amount of working time objectified in the study, combined with the lack of sleep. On the other hand, the job profile of this occupational group is very complex and associated with high, often cumulative, work requirements [34]. The periodically higher stress level was also evident in this study due to the significantly higher maximum working hours among the officers compared to the other occupational groups.

The average total energy expenditure of approx. 3400 kcal objectified in this study indicates that the level of job-related physical exertion for seafarers is still high today, with significant differences between the occupational groups. The subjectively high physical load, especially that of the deck ratings, corresponds to the objectively proven high number of steps and the higher energy expenditure of this occupational group during working hours. High levels of physical effort are required of the deck ratings, especially in stabilizing the containers with heavy iron rods. However, loading and unloading or repair and maintenance work in the deck area are also associated with physical stress for this group [35].

A limitation of the present study is that it covers only seafarers on container ships in the North/Baltic Sea area including the English Channel (i.e., so-called national trade). Thus, no statements can be made about the strain of seafarers on other vessel types or on ships operating in other areas. In addition, the stress load is likely to be different on board of oceangoing vessels that spend much less time in ports and operate less often in crowded ship lanes. Furthermore, acute stress due to environmental influences such as vibration, ship movement, or different climate zones may increase the strain parameters of the seafarers, but were also not the subject of the present investigation. Altogether, this study represents a relatively extensive on-board investigation that includes objective, biometric and subjective parameters, taking into account the working and rest times on board. Although the presence of scientific researchers during the onboard investigation could have had an influence on the crews’ behaviour and may have promoted socially desired answers, this is the only way to verify the working and rest hours as well as the strain parameters of the crew members in an objective manner. 

## 5. Conclusions

The present study is a pioneering study providing the first step to developing job exposure matrices in seafaring. For prevention, a tailor-made health promotion programme on board should be developed with, for example, guidance on a professionally adapted sport programme [20,36]. Given the long working hours over many months on board, a further preventive measure would be a shortened duration of stay on the vessels for the crew members. Additionally, in light of the frequent port turnovers, a reduction in the number of terminals called at for cargo handling through better scheduling in the port would allow for more relaxation time for seafarers. Moreover, there is a high overall need to conduct further studies to objectify seafarers’ psychophysical strain on board and to develop occupational group-specific health promotion programmes.

## Figures and Tables

**Table 1 ijerph-16-01153-t001:** Description of the study sample.

		Occupational Groups			
	Total Sample	Nautical Officers	Deck Ratings	Engine Room Personnel	*p*
**Ranks**, *n*		22 captains, 45 nautical officers	146 ratings on deck, 12 electricians	21 chief engineers, 34 technical officers, 43 ratings in the engine room	
**Number**	323	67	158	98	
**Age**, mean (SD)	38.3 (11.8)	39.2 (10.5)	36.3 (11.1)	41.0 (13.3)	<0.012 ^a^
**Origin**, *n* (%)					<0.001 ^b^
Europeans	155 (48.0%)	56 (83.6%)	38 (24.1%)	61 (62.2%)	
Southeast Asians	168 (52.0%)	11 (16.4%)	120 (75.9%)	37 (37.8%)	

^a^ Kruskal–Wallis test; ^b^ Chi^2^ test.

**Table 2 ijerph-16-01153-t002:** Subjective stress on the current ship depending on the occupational groups.

		Occupational Groups			
	Total Sample (323)	Nautical Officers (67)	Deck Ratings (158)	Engine room Personnel (98)	*p*
**Subjective stress, *n* (%)**					
physical	207 (64.1%)	18 (26.9%)	118 (74.7%)	71 (72.4%)	<0.001 ^b^
mental	209 (64.7%)	58 (86.6%)	90 (57.0%)	61 (62.2%)	<0.001 ^b^
**Working hours, mean (SD)**					
average	9.5 (1.5)	10.3 (1.8)	9.4 (1.5)	9.2 (1.2)	<0.001 ^a^
maximal	14.0 (3.3)	15.2 (3.3)	13.1 (2.4)	14.7 (4.0)	<0.001 ^a^
Sleep deficit, *n* (%)	176 (54.5%)	45 (67.2%)	89 (56.3%)	42 (42.9%)	0.007 ^b^
**Interruptions of sleep per 24 h ^c^, *n* (%)**					0.062 ^b^
<twice	228 (70.6%)	40 (59.7%)	113 (71.5%)	75 (76.5%)	
≥twice	95 (29.4%)	27 (40.3%)	45 (28.5%)	23 (23.5%)	

^a^ Kruskal–Wallis-test; ^b^ Chi^2^ test; ^c^ average in 24 h.

**Table 3 ijerph-16-01153-t003:** Activity data on the current ship per 24 h according to daily log and armband monitor.

		Occupational Groups			
	Total Sample (323)	Nautical Officers (67)	Deck Ratings (158)	Engine Room Personnel (98)	*p* ^a^
**Average duration of work and rest periods per 24 h**					
According to daily log h (SD) (% of the cruise)					
Working hours	9.3 (0.9) (39.1%)	10.9 (0.2) (45.6%)	9.1 (0.3) (38.0%)	8.5 (0.2) (35.3%)	<0.001
Leisure time	6.8 (0.5) (28.3%)	6.3 (0.2) (26.5%)	6.6 (0.3) (27.6%)	7.4 (0.5) (31.4%)	<0.001
Lying/sleeping time	7.8 (0.7) (32.3%)	6.6 (0.2) (27.3%)	8.2 (0.3) (34.2%)	8.1 (0.3) (32.7%)	<0.001
Sport	0.1 (0.01) (0.4%)	0.02 (0.02) (0.2%)	0.01 (0.01) (0.2%)	0.1 (0.01) (0.5%)	0.163
According to armband monitor ^b^, h (SD)					
Lying time	6.1 (0.6)	4.9 (0.1)	6.4 (0.1)	6.4 (0.1)	<0.001
Sleeping time	5.0 (0.5)	4.7 (0.1)	5.0 (0.2)	5.4 (0.3)	<0.001
Number of steps	12475 (2182)	9741 (1154)	14334 (1129)	11344 (871)	<0.001

^a^ Kruskal–Wallis-test; ^b^ 92.0% duration of stay within the period of observation (completed daily log).

**Table 4 ijerph-16-01153-t004:** Strain during the voyages depending on the occupational groups.

		Occupational Groups			
	Total Sample (236)	Nautical Officers (67)	Deck Rating (118)	Engine Room Personal (57)	*p* ^a^
**Biometric data according to the armband monitor,** estimated marginal means (SE)					
**TEE ^b^** in kcal	3394 (151)	2880 (96)	3563 (63)	3389 (84)	**<0.001** ^e^
**WEE ^c^** in kcal	754 (18)	568 (23)	801 (15)	777 (20)	**<0.001** ^e^
**METs ^d^** pro Tag	1.8 (0.1)	1.5 (0.1)	1.9 (0.1)	1.7 (0.1)	**<0.001** ^e^
**Biometric data according to the Polar watch,** estimated marginal means (SE)					
**Heart rate, min^−1^**	81.6 (1.1)	80.7 (1.3)	80.8 (0.9)	82.2 (1.2)	0.604
**% Heart rate ^f^ > 110 beats min^−1^**	12.3%	7.6%	15.3%	14.1%	0.415
**SDNN ^g^**, ms	17.7 (0.7)	16.0 (0.8)	18.8 (0.6)	17.8 (0.8)	**0.016**

^a^ Mixed Model; ^b^ TEE (“total energy expenditure”): Total energy expenditure per day; ^c^ WEE (“work-related energy expenditure”): Energy expenditure based on a four-hour work period; ^d^ METs: Metabolic equivalent of tasks; ^e^ adjusted for age and body mass index; ^f^ percentage of time above the heart rate > 110 min^−1^ based on a four-hour shift; ^g^ SDNN: standard deviation of normal to normal R-R intervals.

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
