# Peer review of "Stress and Strain among Seafarers Related to the Occupational Groups"

_ijerph, 2019, doi:10.3390/ijerph16071153_

Round 1

Reviewer 1 Report

The best study i ever seen on the occupational workload representing such risk factors for chronic diseases and safe managing of the ship. There is a strong need to prevent these risk factors as pointed out by the authors..

As I have already recommended, this study is extremely important to develop better health and safety for the seafarers. For the authors: this is a case study, with different job categories and work places onboard that serve as the controls for each other. We are very well aware that this is not a causal study that can prove that the exposures are as described, but this is the first study ever that describes these exposures in details. Of course it are is needed to go on and to make more studies in detail based on this first pioneering study. This study is the first step to develop job exposure matrices in seafaring. The development of job exposure matrices is is in line with the actual most advanced research standards in these days.  The job exposure matrices are needed of two reasons: 1. for the systematic improvement of the working and living conditions on board and 2. for developing job exposure indexes serving as proxy exposure indexes for health register studies.

Author Response

We thank the reviewer for the very kind remarks and fully agree with the statements. Particularly the description of our study as a pioneering study providing the first step to developing job exposure matrices in seafaring has been added to the revised manuscript.

Reviewer 2 Report

Major points

1. The authors mentioned the relationship between stress parameter and biometric data in Abstract, Results and Discussion. But the statistical data including methods was not shown. These data should be shown. Instead, Table 2 to Table 4 could be combined.

2. The mixed models were used. However, I could not understand it well. Is the description in Line 111-115 for the mixed model?

3. The conclusions (Line 261-267) should mentioned summary of the paper rather than implication and further research.

Minor points

4. The statistical software should be mentioned. “Tab. *” in the text should be “Table *”

5. Line 79: The measurement of psychophysical stress was not clear.

6. Table 4: MET and SDNN should be explained in the footnotes.

Author Response

 The Statistics Section has been improved in the revised manuscript: firstly, we have included the information that statistical analysis was performed with SPSS (version 25, IBM Corporation). Secondly, the definition of the mixed model has been specified. Thirdly, the fact that the Spearman’s correlation was conducted to assess the dependence between the rankings of two variables has been explained, and fourthly, the correlation values of the significant differences have been added.

The tables 2 to 4 have different focus (subjective vs. objective; activity related vs. objective biometric data) and are linked to corresponding different subsections. Thus, we would prefer to continue separating these tables.

To our understanding, the conclusion should focus on the consequences that are based on the study findings. Thus, we have deleted the link about the 10,000 steps per day recommended by the WHO and have added some ideas on how to improve the seafarers’ situation on board (as requested by reviewer 1).

All mentioned “minor points” have been addressed accordingly.

Reviewer 3 Report

The paper could be imp[roved with comments in two areas. In general, the representation of the officers in this study seems high compared to crew- in general what is the ratio of number of officers to number of crew. Secondly, this report may not be generalizable since mostly done with ships in local commerce with likely frequent port calls. Might the results be different if the ship crews studied were of longer ocean voyages with much less time spent in ports or crowded ship lanes. line 48-needs clarification regarding "highly frequented river passage". Discussion about how to improve things beyong the one thought would be welcome as well.

Author Response

Normally, the crews on small container ships consist of 6 officers (master, 1st and 2nd nautical officer, chief, 2nd and 3rd technical officer) and 10 ratings resulting in a usual ratio of 37.5% (6/16) officers and 62.5% (10/16) ratings. This corresponds to the ratio in our study.

Yes, we agree that our findings are related to container ships operating in the national trade with frequent port calls. Thus, the stress load is likely to be different on board of ocean-going vessels that spend much less time in ports and operate less often in crowded ship lanes. This aspect has been added as a limitation of the study.

The term “highly frequented river passage” has been rephrased to “river passage with crowded ship lanes”.

The discussion has been enriched by some additional recommendations that seem to be suitable to improve the working and living situation on board:

·        A tailor-made health promotion programme on board

·        A shortened duration of stay on the vessels for the crew members

·        A reduction in the number of terminals for cargo handling

·        The conducting of further studies to objectify seafarers’ psychophysical strain on board.

Round 2

Reviewer 2 Report

The manuscript was revised according to the reviewers'  comments.